# Evaluation of Sample Preparation Methods for the Analysis of Reef-Building Corals Using ^1^H-NMR-Based Metabolomics

**DOI:** 10.3390/metabo9020032

**Published:** 2019-02-13

**Authors:** Erik R. Andersson, Rusty D. Day, Julie M. Loewenstein, Cheryl M. Woodley, Tracey B. Schock

**Affiliations:** 1Grice Marine Laboratory, Department of Biology, College of Charleston, 205 Fort Johnson Rd., Charleston, SC 29412, USA; loewensteinjm@g.cofc.edu; 2Hollings Marine Laboratory, Chemical Science Division, National Institute of Standards and Technology, Charleston, SC 29412, USA; rusty.day@manta-online.org (R.D.D.); tracey.schock@nist.gov (T.B.S.); 3Marine Science and Nautical Training Academy (MANTA), 520 Folly Rd., Charleston, SC 29412, USA; 4Hollings Marine Laboratory, National Ocean Service, National Oceanic and Atmospheric Administration, Charleston, SC 29412, USA; cheryl.woodley@noaa.gov

**Keywords:** metabolomics, method, NMR spectroscopy, sample preparation, coral

## Abstract

The field of metabolomics generally lacks standardized methods for the preparation of samples prior to analysis. This is especially true for metabolomics of reef-building corals, where the handful of studies that were published employ a range of sample preparation protocols. The utilization of metabolomics may prove essential in understanding coral biology in the face of increasing environmental threats, and an optimized method for preparing coral samples for metabolomics analysis would aid this cause. The current study evaluates three important steps during sample processing of stony corals: (i) metabolite extraction, (ii) metabolism preservation, and (iii) subsampling. Results indicate that a modified Bligh and Dyer extraction is more reproducible across multiple coral species compared to methyl *tert*-butyl ether and methanol extractions, while a methanol extraction is superior for feature detection. Additionally, few differences were detected between spectra from frozen or lyophilized coral samples. Finally, extraction of entire coral nubbins increased feature detection, but decreased throughput and was more susceptible to subsampling error compared to a novel tissue powder subsampling method. Overall, we recommend the use of a modified Bligh and Dyer extraction, lyophilized samples, and the analysis of brushed tissue powder for the preparation of reef-building coral samples for ^1^H NMR metabolomics.

## 1. Introduction

The ecological and economic importance of coral reefs is well described, as these ecosystems provide a majority of the ocean’s biodiversity and productivity [1,2]. However, the health of coral reef ecosystems declined in recent decades due to increasing global- and local-scale stressors [3,4]. Increasing atmospheric carbon dioxide levels leading to ocean acidification and increasing oceanic temperatures universally threaten coral reef ecosystems worldwide [5]. At a local level, anthropogenic chemical pollution, sedimentation, and eutrophication may also negatively affect coral reef health and resilience [6,7]. The culmination of these various anthropogenic stressors seems to compound and further threaten coral reef health. For example, coral diseases concurrently increased in number and prevalence in recent decades, and are now considered an important threat to coral reef ecosystems worldwide [8,9,10]. Decisions related to coral reef management depend on the integration of socio-economic pressures and the best biological information available [11,12,13]. Therefore, high priority is currently placed on research efforts toward monitoring and assessing coral reef ecosystem health, as evidenced by the large number of recent studies evaluating various aspects of reef health across the globe (e.g., References [14,15,16,17]). Additionally, further understanding the complex biological processes underpinning coral stress responses will be imperative for managing coral reef ecosystems in the face of mounting anthropogenic pressures.

The ability to investigate biological questions at the molecular level dramatically improved with the development of -omics techniques arising in the mid 1990s, which transformed biological research into a data-rich discipline [18]. Consequently, the use of these molecular tools provided new insights into coral biology and stress response by studying various parts of the coral “ome”, such as the genome, transcriptome, microbiome, and proteome (e.g., References [19,20,21,22,23]). Metabolomics is the most recent development among the -omics tools, defined as the study of low-molecular-weight compounds from cells, tissues, or biological fluids under specified physiological conditions [24]. Here, we use the term to refer specifically to the study of soluble polar metabolites, as these compounds are the primary focus of the current study. These compounds are metabolic intermediates and products, which reflect downstream effects of gene expression and enzymatic activity; therefore, the metabolome is considered to be more sensitive to biochemical activity than the transcriptome or proteome [25,26,27]. This close link to biochemical phenotype makes metabolomics ideal for careful monitoring of organism response to perturbation [26,28], and metabolomics were successfully applied to a variety of aquatic organisms including fishes, invertebrates, and microbes [27,29,30,31]. For these reasons, metabolomics holds immense potential in a wide range of aquatic fields, with applications in ecology, toxicology, and aquaculture, among others [27,32,33,34]. These characteristics suggest that the use of metabolomics has the potential to advance the understanding of coral response to stressors, yet the application of metabolomics to reef-building corals remains widely unexplored [35]. To date, a limited number of metabolomics studies of stony corals were published, with only a handful of those being NMR-based (Appendix A, Appendix A) [35,36,37,38,39,40,41,42,43,44]. 

As a young field, metabolomics suffers from a lack of standardized methods, which is accentuated by the fact that the goals of a given metabolomics study will necessarily determine the optimal methods [24,45]. For example, whether a metabolomics study focuses on quantifying a small set of predetermined compounds (targeted [46]) or aims to detect as many metabolites as possible to evaluate the composition of entire metabolic profiles (untargeted [47]) will largely determine appropriate protocols. Additionally, subtler considerations including the biological matrix and analytical instrumentation may also influence desired protocols [24]. The lack of standardized methods holds especially true for metabolomics of reef-building corals, where the common practices in metabolomics should not be applied automatically due to the unique biological matrix presented by these organisms. Reef-building coral samples do not solely consist of the coral animal itself, but also their symbiotic algae (zooxanthellae), complex assemblages of associated microbes, and endolithic algae, collectively called the coral holobiont [48]. Additionally, the coral soft tissues are contiguous with the coral calcium carbonate (CaCO_3_) skeleton, and, to our knowledge, no procedures appropriate for environmental metabolomics currently exist to separate the coral tissue and skeleton. The entire coral holobiont (including the skeleton) must, therefore, be analyzed, often at the expense of elucidating individual metabolite origins from within the holobiont. This results in a relatively complex biological matrix that presents unique challenges during sample processing and interpretation.

Generally, metabolomics workflows include sample collection, sample preparation, chemical analysis/detection, data processing, statistical analyses, and finally biological interpretations [24]. Depending on the sample and study type, sample preparation may require some combination of intermediate steps such as rapid metabolism quenching, sample storage, metabolism preservation during handling, subsampling, and metabolite extraction [24,36,49]. The choice and execution of these steps must be carefully considered in order to ensure the quality of metabolomics results [24,45]. At least one study attempted to establish standard sample preparation procedures for metabolomics of stony corals [36]; however, coral metabolomics studies to date utilized a wide combination of sample processing methods (Appendix A, Appendix A). Therefore, the field would undoubtedly benefit from additional comparisons of possible methods for important sample preparation steps applicable to reef-building coral samples destined for metabolomics analysis. 

Here, we provide empirical and conceptual comparisons of possible methods for three important steps during sample preparation of stony coral samples for proton nuclear magnetic resonance (^1^H NMR)-based metabolomics. These steps, in the order in which the experiments were conducted, are (i) metabolite extraction, where samples are submerged and agitated in particular solvent mixtures in order to extract a broad survey of polar metabolites for analysis; (ii) metabolism preservation, where the time-specific metabolism is maintained after initial quenching through the early steps of sample processing (e.g., aliquoting, weighing) to prevent the introduction of additional variation prior to chemical extraction; and (iii) subsampling methods, where samples are manipulated into a form conducive to chemical analysis. These results guided the development of a sample preparation workflow that we recommend for untargeted NMR metabolomics studies investigating fundamental coral biology and/or molecular impacts of perturbations (i.e., stressors, disease) to reef-building corals.

## 2. Results and Discussion

### 2.1. Extraction Method Comparison

The metabolite extraction methods tested here were all used in previous metabolomics studies to varying degrees (e.g., References [36,50,51,52,53]) and each method possesses specific considerations, making them differentially useful depending on the sample type and the goals of a particular study. Here, the volume of extraction solvents (Appendix A, Appendix A) and the extraction protocols were kept relatively consistent across methods so that the effects of solvent choice and solvent ratios could primarily be investigated. The Bligh and Dyer extraction method was originally developed to separate and remove lipids from a biological sample [54], and is generally considered to be the gold standard for biphasic extractions in metabolomics studies [51]. The methyl *tert*-butyl ether (MTBE) extraction method was later developed to provide faster and cleaner lipid extraction by replacing chloroform with a less dense non-polar solvent, MTBE. Methanol extraction was proposed as a method to maximize metabolite extraction for non-targeted metabolomics of stony corals [36]. Most coral metabolomics studies to date used a methanol extraction method [35,36,39,40,41,42,43,44], and none used the Bligh and Dyer or the MTBE methods, despite their widespread use in the metabolomics field [53,55]. 

Bligh and Dyer extraction results in a biphasic solvent system. The top, polar fraction comprising methanol and water contains hydrophilic metabolites, while the bottom, non-polar fraction comprising chloroform primarily contains lipids. An insoluble protein plug sits in between the two fractions and contains both proteins and nucleic acids from the sample, and an additional CaCO_3_ skeletal plug unique to stony corals lies at the very bottom (Figure 1). The MTBE extraction also results in a biphasic solvent system; however, in this system, the polar fraction falls below the less dense, non-polar MTBE fraction. Here, the insoluble protein plug and CaCO_3_ skeletal plug both lie below the aqueous and organic fractions (Figure 1). The methanol extraction notably does not incorporate a non-polar solvent, and results in a single extract fraction directly above the pelleted coral/skeleton material (Figure 1).

Visual assessment of ^1^H NMR spectra reveals similar peak coverage and intensities for all three extraction methods for both *Acropora cervicornis* and *Orbicella faveolata*, with slightly more detectable features and higher peak intensities in spectra resulting from the methanol extraction method (especially for *A. cervicornis*) (Appendix A, Appendix A; Table 1). Multivariate analysis with principal component analysis (PCA) containing all samples shows grouping by species along the first principal component (PC 1, 84.5% explained variance (EV)) and by extraction method along PC 2 (11.1% EV) (Appendix A, Appendix A), indicating that the variation in metabolic profile between species is much greater than the variation between the different extraction methods.

PCA score plots of PC 1 vs. PC 2 for the *A. cervicornis* samples only (combined 92.2% EV) and for the *O. faveolata* samples only (combined 90.8% EV) show the Bligh and Dyer samples consistently grouping together with relatively low variation (Figure 2). Small spectral relative standard deviation (spectral %RSD, see Section 3 for description) values of 7.82 and 5.85 for the respective coral species further support this observation. The MTBE samples appear highly reproducible for *A. cervicornis* (spectral %RSD = 6.45) but display considerable variation for *O. faveolata* (spectral %RSD = 10.55) (Figure 2). Additionally, MTBE extracts a more different metabolite profile than the Bligh and Dyer and methanol methods (Figure 2, *O. faveolata*: PC 1, *A. cervicornis*: PC 2). The methanol samples display the least consistent grouping on the PCA score plots, particularly for *A. cervicornis* (spectral %RSD = 13.30), where all four samples visibly separate along PC 1 (Figure 2). It is unclear why the MTBE and methanol extractions vary in their grouping patterns and spectral %RSD values between species. This observation might indicate that extraction solvents fluctuate in their effectiveness across different coral species, possibly stemming from inter-species variation in tissue composition, and warrants the comparison of metabolite extraction solvents over a wider selection of coral species.

Overall, these results indicate that the methanol was less reproducible than both the Bligh and Dyer and the MTBE extraction methods. This may be explained by the biphasic nature of the Bligh and Dyer and the MTBE methods, which, unlike the methanol extraction, isolate the polar metabolites desired for metabolomics from the lipids and cellular protein debris. In the vein of maintaining a high-throughput extraction, the solvent volume per mass of homogenate may be too low and, thus, results in a less efficient extraction, causing variability in the polar lipids which overwhelm the NMR spectra. Introduction of a non-polar solvent to partition the lipids from the polar systems creates a more efficient and reproducible extraction. Reproducibility is paramount for high-quality metabolomics studies, in order to understand and minimize variation from sources other than the treatment of interest (i.e., technical and inter-sample variation) and ensure accurate interpretation of results [57]. Our results regarding reproducibility of different solvents support previous findings in the metabolomics literature. For example, previous studies demonstrate that methanol:water and methanol:chloroform:water extractions generally result in spectral %RSD values around or below 10% for NMR-based studies [57], and that biphasic extractions outperform reproducibility of single-phase methanol extractions in fish tissues [51]. 

Both the Bligh and Dyer and MTBE methods provide the ability to collect and save the different extraction phases for separate multi-omics analyses. For example, in addition to the desired polar fraction, the non-polar fraction and protein plug can be saved for future lipid and protein analyses, respectively. This approach allows for multiple unique molecular analyses of a single extract, where the integrated data combine to provide a systems biology perspective of the samples. 

Current coral processing methods lack the ability to separate the coral tissues from the coral skeleton without compromising the metabolic state of the tissues. This complex biological matrix presents unique challenges when processing and extracting stony coral samples for metabolomics analysis, as extraction procedures are complicated by the large amounts of non-extractable CaCO_3_. This matrix makes standardization of tissue:solvent ratios during extraction difficult, a challenge that persists in the current study. However, because extraction efficiency is not a sole/primary factor in the selection of preferred methods, we contend that any biases introduced due to discrepancies in tissue:solvent ratio do not invalidate the overall evaluation of compared methods. Although the presence of the CaCO_3_ does not prominently alter the final spectra, this material increases difficulty of extraction tube transfer steps and, therefore, increases the possibility of sample loss, adding more technical variation between samples during extraction. The methanol method used fewer solvents and, therefore, had fewer tube transfer steps compared to the two biphasic extractions, making this method less susceptible to irreproducibility from this source.

Although the simpler protocol and higher peak intensities afforded by the methanol may be beneficial for certain applications, the lower overall reproducibility and inability to easily preserve materials for complementary analytical analyses prevent this extraction method from achieving optimal status under the goals of the current study. Practicality and reproducibility are comparable and relatively strong for both the Bligh and Dyer and MTBE extractions; therefore, both of these methods are generally considered desirable here. However, the polar metabolite fraction constitutes the top-most layer when using the Bligh and Dyer extraction, while the polar metabolites are on the bottom when using the MTBE extraction (Figure 1). This makes collection of the polar fraction using the Bligh and Dyer method easier and less susceptible to contamination from the other layers. Additionally, the MTBE method results in both the protein plug and undesirable coral skeleton lying at the bottom of the collection tube, which limits the ability to collect protein material for multi-omics analysis. Finally, the MTBE extraction spectra contain contamination from MTBE solvent around 3.20 ppm (Appendix A, Appendix A), a region of the spectrum often populated by important features. In contrast, the chloroform contamination from the Bligh and Dyer extraction occurs at 7.68 ppm (Appendix A, Appendix A), a relatively sparse area of the spectrum. For these reasons, we recommend the Bligh and Dyer extraction method as the preferred metabolite extraction method.

### 2.2. Metabolism Preservation Comparison

Maintaining the in vivo metabolism of samples throughout sample handling is paramount to metabolomics experiments [24,51]. Freezing samples using liquid nitrogen (LN_2_) is a common method for initially quenching metabolism [24,28,51,55], and maintaining freezing temperatures during sample handling aids in minimizing alterations to the metabolic state of interest at time of collection [24]. However, preserving a frozen state while processing and preparing samples for extraction comes with challenges. Any time spent at room temperature risks tissue thawing and subsequent metabolic activity. Therefore, frozen samples are often handled in low-temperature environments as can be provided by cryocarts. Sample metabolism may also be preserved via lyophilization, which theoretically removes all water from the samples, thereby quenching their metabolism [58]. One concern with sample lyophilization is the possibility that the process itself may alter the metabolic state of interest; for example, this process may remove volatile compounds from the samples [36]. Most coral metabolomics studies used lyophilization in order to preserve metabolism during sample processing [35,36,40,42], although a few worked with frozen samples [41,44]. 

Comparison of ^1^H NMR spectra from the frozen and lyophilized homogenates reveals metabolic profiles that are visually similar between the two treatments (Appendix A, Appendix A). Average spectral intensity and the number of detectable features are slightly higher for lyophilized compared to frozen coral (Table 1). Despite the lack of overtly noticeable differences in the raw data, on the PCA score plot of PC 1 vs. PC 2 (combined 90.3% EV), when comparing frozen and lyophilized samples, the two treatments separate distinctly along PC 1 (Figure 3). However, samples group together for both treatments and are highly reproducible with spectral %RSD values of 4.95 and 5.79 for frozen and lyophilized samples, respectively (Figure 3).

It does appear that lyophilizing the coral samples has some effect on the metabolic profile, as indicated by the separation of the two treatment groups along PC 1 of the PCA scores plot. However, the low intra-treatment variation, evidenced by the low variability of lyophilized samples on PC 1 and the spectral %RSD value (Figure 3), indicates that the metabolic discrepancies between the methodologies are consistent. Furthermore, our metrics evaluating spectral coverage and intensity fail to discriminate between frozen and lyophilized samples. Overall, these results suggest that, although lyophilization may alter the metabolic profile of these coral samples, any alterations are relatively subtle and are consistent between samples and, therefore, do not compromise profile integrity.

Handling samples in a cryocart cooled by LN_2_ helps prevent thawing, but also limits the ability to inspect and manipulate samples, as they must be kept within the cart to prevent thawing. Working in a cryocart also requires the use of cryogenic gloves, which further lower user dexterity. Unlike frozen samples, lyophilized samples can be handled at room temperature, which allows for increased dexterity and clear visual inspection of the samples during processing. However, any time spent working with the samples risks re-hydration via moisture from the air, which may alter the metabolic state of interest. Therefore, lyophilized samples should be handled in a biological safety hood where processing time and exposure to air are minimized, and stored in tightly closed vials at −80 °C when not actively being handled. The ability to work more freely with lyophilized samples as described above provides the benefit of increasing the precision and throughput of sample preparation protocols. Additionally, the reduced ability to examine individual specimens while using a cryocart limits the potential to collect high-quality metadata, identify and avoid biological or methodological contamination, and target desired morphological structures within individual samples. This may be particularly detrimental when studying environmental samples, as any of these steps can make or break interpretation of metabolomics results. Finally, metabolite extraction becomes simpler when working with lyophilized samples, as moisture content of the tissue does not need to be considered when maintaining the correct solvent ratios for proper phase separation. 

Although metabolic profiles are distinguishable using multivariate statistical methods, both treatments remain highly reproducible and do not show dramatic visual discrepancies in the raw data. Therefore, logistical considerations are the most influential factors in determining the optimal method for metabolism preservation. Because of the practical improvements to the sample preparation laboratory work associated with the use of lyophilized samples, we recommend this as the preferred metabolism preservation method.

### 2.3. Subsampling Method Comparison

Subsampling methods to date are inconsistent across metabolomics studies of stony corals. Often, small nubbins are removed from the larger coral sample and extracted whole [35,36,39,40,42,43], or the entire coral sample (tissue plus underlying skeleton) is homogenized and extracted [38]. However, the use of nubbins presents standardization and normalization challenges, as it can be difficult to accurately estimate the amount of extractable tissue present in a given nubbin. Coral homogenates are largely contaminated by the coral skeleton creating practical difficulties during metabolite extraction. Both of these approaches indiscriminately incorporate metabolites from the entire coral holobiont (coral, zooxanthellae, microbes, endolithic organisms (likely limited in nubbin extractions)) into their analyses. Some studies attempted to separate the coral tissue from its skeleton prior to analysis, for example, by airbrushing the coral tissue away from the skeleton [41,44]. However, this method involves allowing the samples to thaw and interact with water before extraction, likely compromising the integrity of the metabolic state of interest. Our nubbin subsampling method closely mirrored coral subsampling methods used in previous metabolomics studies (e.g., Reference [36]), while our tissue powder method provided a novel approach for preparing coral samples for NMR metabolomics analysis.

Evaluation of the ^1^H NMR spectra indicates that the nubbin subsampling method results in much higher peak intensities (order of magnitude), as well as a near 50% increase in the number of detectable features compared to the tissue powder subsampling method (Appendix A, Appendix A; Table 1). A PCA comparing both subsampling methods (PC 1 vs. PC 2, combined 96.2% EV) displays the two methods separated clearly along PC 1, likely reflecting the observable increase in spectral coverage and intensity of the nubbin method (Figure 4). However, the tissue powder samples formed a tight cluster (spectral %RSD = 5.72) while the nubbin samples spread out across PC 2 (spectral %RSD = 18.40) (Figure 4). This suggests that the tissue powder method produces a highly reproducible ^1^H NMR spectrum relative to the more variable nubbin method.

Fundamental differences in the protocols of these two subsampling methods may help explain these findings. The relatively larger nubbins (~1.5 g each) undoubtedly contain more extractable tissues compared to the tissue powder (~20 mg each), which likely drives the disparity in spectral coverage and intensity between the two methods. Additionally, the tissue powder material is combined into a homogeneous powder and then aliquoted into what are essentially technical replicates, whereas producing identical nubbins from a given coral sample is not feasible, making repeat analyses or technical replication of a sample challenging. For this reason, the nubbin samples here also reflect biological variability within the colony in addition to methodological variability. Although these results do not represent an entirely standard comparison of profile coverage and reproducibility between tissue powder and nubbin methods, they demonstrate that the best available practices of the tissue powder method produce less material but a more reproducible extract than the best available practices of the nubbin method. Therefore, the ability to aliquot technical replicates is currently considered a benefit of the tissue powder method. 

The nubbins themselves are relatively large and need to be completely submerged during metabolite extraction, and, therefore, higher total volumes of solvent are necessary for extraction of the nubbins. This larger solvent volume makes nubbin extraction protocols more cumbersome and reduces throughput compared to the smaller volumes of the tissue powder method. High-throughput methods are essential to metabolomics to allow as many biological replicates as possible during metabolite extraction, while minimizing the amount of time that samples sit between extraction steps (possibly altering the desired metabolic state). Even if the maximum size of coral nubbins to be analyzed is limited so as to minimize solvent volume, the upper limit of nubbin sizes that can reasonably be extracted is still often much smaller than the size of the original coral samples. Because a given coral sample is likely to contain spatial variations in morphology, physiology, and metabolism [59], a relatively small nubbin removed from a larger coral sample would likely be vulnerable to subsampling error where some of this spatial variability is not captured. In contrast to our results, one study found little difference in the reproducibility of ^1^H NMR profiles between intra-colony nubbin samples and technical replicates of *P. compressa* [35]. However, spectral %RSD values reported for technical variation in the aforementioned study (14.2) were notably higher than spectral %RSD values from our tissue powder technical replicates (5.7), possibly as a result of their use of homogenized samples (high CaCO_3_:tissue ratio) compared to our tissue powder method (lower CaCO_3_:tissue ratio (Appendix A, Appendix A)).

In contrast, the tissue powder subsampling method allows for targeted integration or isolation of the inherent spatial variability across a given coral sample, producing a material that is theoretically representative of the sample or area of interest. Furthermore, tissue powder from the same sample provides a more consistent metabolic profile than nubbins from the same sample (Figure 4). The material collected using the tissue powder method is easily separated into technical replicates or stored at −80 °C, allowing for repeat examination of the same material using NMR metabolomics or other analytical endpoints, and feature detection capability can be tuned by adjusting the amount of tissue powder to be extracted. Although the tissue powder method does not completely remove the coral tissue from the skeleton, it greatly reduces the mass of undesirable skeleton in the sample compared to methods such as whole-fragment homogenization. Additionally, the tissue powder method should reduce contamination from endolithic organisms deep within the CaCO_3_ structure, which are not excluded from extraction with a homogenized fragment, and may be extracted with the nubbin subsampling method as well. However, here, we are still unable to determine the origin of the remaining metabolites from within the holobiont. Going forward, pairing metabolomics data with other complementary -omics data may help parse these metabolite assignments (e.g., Reference [42]). Finally, the tissue powder subsampling method requires more steps during sample processing compared to the nubbin method, including extensive brushing of the samples. These additional steps create the risk of altering the metabolic state of interest during sample processing; however, our results suggest analyst consistency with the tissue powder method. 

Overall, the nubbin subsampling method generates higher peak intensities in the ^1^H NMR spectra, while the tissue powder method provides greater reproducibility in addition to many logistical advantages. These include the ability to collect material more representative of the entire coral sample, and to store extra material for technical replication or for separate analyses. For the purpose of this study, it was determined that the advantages of the tissue powder method outweigh the spectral intensity produced by the nubbin method and we, therefore, recommend the tissue powder technique as the preferred subsampling method. 

### 2.4. Selection of Final Recommended Workflow

The final recommended workflow for sample preparation of reef-building coral samples for ^1^H NMR metabolomics was determined based on the relative benefits of the compared methods for metabolite extraction, metabolism preservation, and subsampling steps. We recommend the following sample preparation for reef-building corals: lyophilization of samples for metabolism preservation during handling, subsampling using the tissue powder method, and metabolite extraction using the Bligh and Dyer method. 

### 2.5. Validation of Recommended Sample Preparation Workflow for Stony Corals

An effective workflow will be able to produce high-quality NMR spectra and be sensitive enough to consistently detect differences in coral samples of differing treatments, such as the healthy (unaffected) and diseased (growth anomaly) *Porites compressa* samples analyzed here, while also keeping variation from methodological sources low. Coral samples processed using the recommended workflow result in ^1^H NMR spectra that are generally similar in their peak coverage and intensities for the growth anomaly and unaffected *P. compressa* (Appendix A, Appendix A). Close inspection of the spectra does not reveal novel features between the growth anomaly and unaffected sample, but many features differ in intensity (Appendix A, Appendix A). Technical replication of the growth anomaly and unaffected samples demonstrates high reproducibility (spectral %RSD values <10) and a separation along PC 1 between treatment groups (Figure 5). The relatively large inter-group (biological) variation along PC 1 (91.2% EV) in comparison with the lower intra-group (methodological) variation along PC 2 (5.5% EV) supports the utility of metabolomics and the selected workflow to study corals. These results indicate that the methods developed here provide high enough resolution to detect relatively subtle differences between diseased vs. healthy coral samples from the same colony with generally similar overall metabolic profiles. 

A direct comparison of ^1^H NMR spectra from healthy *P. compressa* using our recommended workflow to those in the literature [35] reveals relatively similar profiles (Figure 6), with slight differences in peak intensities likely due to the choice of subsampling protocols (Appendix A, Appendix A) leading to differences in the amount of extracted sample between the two. This favorable comparison supports the assertion that the developed method effectively captures the coral metabolic profiles, and highlights the ability of different coral sample preparation methods to produce serviceable metabolomics data. However, our workflow provides benefits in reproducibility and logistical capacities such as sample handling, spatial variability, and multi-omics potential, which represent an optimization of sample preparation methods for coral samples used in a manner for direct comparison. Therefore, overall, these results validate the recommended workflow for experimental investigation of field-collected stony coral samples. Because the current study only utilizes three species of reef-building coral, extrapolation of these results to other species should be done with some caution. However, the species studied here intentionally include a variety of different skeletal (perforate and imperforate) and morphological (branching and massive) chariteristics in order to provide as representative of a sampling as possible given the limited species count. 

## 3. Materials and Methods

### 3.1. Sample Collection

Coral samples of two species, *A. cervicornis* and *O. faveolata*, were collected from the National Oceanic and Atmospheric Administration (NOAA) coral culture facility (Permit FKNMS-2016-021 issued to Dr. Cheryl Woodley). Approximately 25 g of coral fragments from each species was removed from their propagation plugs using stainless-steel bone-cutting forceps. Collected fragments were immediately placed into Teflon bags and frozen in LN_2_ to preserve the metabolic state of the corals. Frozen fragments were then cryohomogenized using a Retsch Cryomill (RETSCH GmbH, Haan Germany) and stored at −80 °C until analysis. These homogenates were used to compare the application of (i) traditional metabolite extraction solvents, and (ii) different metabolite preservation techniques to stony corals for NMR metabolomics analysis.

Additionally, coral samples were collected from living *P. compressa* colonies from Kaneohe Bay, Oahu, Hawaii at Coconut Island (State of Hawaii DAR (Division of Aquatic Resources) Special Activity Permit (SAP) 2011-1). Samples were collected within a 1-h period from depths of less than 3 m using a stainless-steel hammer and chisel to extract skeletal biopsies approximately 4 cm in diameter from colonies. Samples were immediately taken to the surface and frozen in LN_2_ to preserve metabolic state. These samples consisted of diseased (growth anomaly) and apparently healthy (unaffected) coral fragments from colonies containing coral growth anomalies as a part of a related study investigating the effects of this disease on corals. One paired (growth anomaly and unaffected) set of these samples was sacrificed for use in the current study. The unaffected sample was used in order to compare use of (iii) different subsampling methods of stony corals for NMR metabolomics analysis, and both the growth anomaly and unaffected samples were used to validate the selected workflow for use on environmental coral samples.

### 3.2. Metrics for Selecting Sample Preparation Methods

Although the goal of a given study will determine which methods are used for each sample preparation step, common metrics for evaluating different methods in metabolomics workflows include reproducibility, feature detection, and logistical considerations [24,49,51]. Here, we considered reproducibility, ^1^H NMR profile coverage and intensity, and multi-omic capability, as well as practical concerns unique to stony coral samples such as dealing with the ever-present coral skeleton. We tested protocols both from previous coral metabolomics studies, as well as novel techniques developed during the current study (Appendix A, Appendix A). Results from each of these categories were considered in combination to determine the preferred method for each of the three sample preparation steps in question, leading to the selection of an optimal workflow for use in untargeted NMR metabolomics coral studies.

### 3.3. Extraction Method Comparison

Three metabolite extraction methods for application to ^1^H NMR metabolomics were tested using the frozen *A. cervicornis* (200 mg (±6 mg)) and *O. faveolata* homogenates (100 mg (±3 mg)): Bligh and Dyer extraction (*n* = 4 per species), MTBE extraction (*n* = 4 per species), and a methanol (Fisher Chemical, HPLC Grade) extraction (*n* = 4 per species) [36,54,60]. Because the two coral species had contrasting morphologies (branching and bouldering), target masses of homogenate for extraction differed by species in order to roughly account for how these morphological differences affected the amount of tissue present in each homogenate. It is important to consider that, when working with stony corals, a certain portion of each sample will comprise the inorganic coral skeleton. Extraction solvent volumes were targeted for a consistent amount of solvent (1000 µL) for each solvent system; however, variations are caused by differing sample moisture content, variations in the mass of tissue/material to be extracted, and desired solvent:volume ratios, and are noted below and in Appendix A (Appendix A). All extraction solvent ratios indicated below are by volume. Samples were extracted in a single batch, which also contained a blank sample for each extraction method to monitor for possible method contaminants. 

#### 3.3.1. Bligh and Dyer Extraction 

A polar solvent system consisting of water (including water content from sample tissues) and methanol (water:methanol ratio of 0.8:2), and a non-polar solvent system consisting of water and chloroform (water:chloroform ratio of 1:2) were used to extract metabolites from the frozen coral homogenates. This resulted in a final chloroform:methanol:water ratio of 2:2:1.8, and total solvent volumes were to 931 µL and 1172 µL for *A. cervicornis* and *O. faveolata* homogenates, respectively. Metabolites were extracted from the frozen coral homogenates using steps modified from Reference [61]. Firstly, the pre-cooled polar solvent system was added to the homogenates in 1.5-mL microcentrifuge tubes and were bath sonicated with ice for 2 min. Next, the homogenate/polar solvent mixture was transferred into a glass vial containing the pre-cooled non-polar solvent system and was vortexed for 60 s. The extracts were then allowed to sit on ice for 10 min before centrifugation at 2000× *g* (standard acceleration due to gravity) at 4 °C for 5 min for phase separation. The polar metabolite-containing fraction was transferred using a glass pipette into a microcentrifuge tube and then dried in a Vacufuge Concentrator 5301 (Eppendorf AG, Hamburg Germany) at room temperature.

#### 3.3.2. MTBE Extraction 

A polar solvent system consisting of water and methanol (water:methanol ratio of 2:6), and a non-polar solvent system consisting of water and chloroform (water:MTBE ratio of 5:20) were used to extract metabolites from the frozen coral homogenates. This resulted in a final MTBE:methanol:water ratio of 20:6:7, with a total solvent volume of 1237 µL. Metabolites were extracted from frozen coral homogenates following the same step-wise protocol as above.

#### 3.3.3. Methanol Extraction 

A 70% (*v*/*v*) methanol solvent system was added to the frozen coral homogenates to a total solvent volume of 1000 µL. This extraction solvent did not include a non-polar solvent system. Metabolites were extracted from frozen coral homogenates by adding the pre-chilled solvents to the homogenates in 1.5-mL microcentrifuge tubes. The homogenate/solvent mixture was then vortexed for 10 s prior to an iced-bath sonication for 5 min. After sonication, extracts were spun at 14,000× *g* at 4 °C for 5 min to pellet coral homogenate. Supernatant was then transferred into a new microcentrifuge tube and dried in a Vacufuge Concentrator 5301 (Eppendorf AG, Hamburg, Germany) at room temperature.

### 3.4. Metabolism Preservation Comparison

Two metabolism preservation techniques were tested using the *O. faveolata* homogenate: frozen and lyophilized methods. Frozen (*n* = 6) and lyophilized (*n* = 6) samples were extracted and analyzed using ^1^H NMR spectroscopy in a single batch which included a blank sample for each sample state to monitor for possible method contaminants. All of the frozen and lyophilized samples were technical replicates of a single *O. faveolata* homogenate as described above.

For both the frozen and lyophilized methods, frozen homogenates (100 mg (±3 mg)) were firstly transferred into pre-weighed 10-mL glass vials using a cryocart in order to maintain freezing temperature. Samples for the frozen method were stored at −80 °C until metabolite extraction, while samples for the lyophilized method were freeze-dried for 24 h at an average pressure below 100 mTorr in pre-chilled glass beakers (Appropriate Technical Resources CT 110 Lyophilizer) prior to storage at −80 °C. Metabolites were extracted from all samples using the Bligh and Dyer extraction methods described above, with the sonication step being replaced by a 20-s vortex. The total solvent volumes were to 903 µL and 1088 µL for frozen and lyophilized homogenates, respectively.

### 3.5. Subsampling Method Comparison

Two subsampling methods were tested using the sacrificial unaffected *P. compressa* sample: nubbin subsampling (*n* = 3) and tissue powder subsampling (*n* = 3). The coral sample was lyophilized using a VirTis Genesis OX Lyophilizer with a Wizard 2.0 Controller for 24 h at an average pressure below 1 Torr and a shelf temperature of 20 °C. The sample was then divided into two coral fragments using a chisel and hammer for use in the two different subsampling methods to be tested (Appendix A, Appendix A). Samples from the two methods were extracted in a single batch, which also contained a blank sample for each of the subsampling methods to monitor for possible method contaminants.

#### 3.5.1. Nubbin Method 

Three small nubbins (1.5 g (±0.045 g)) were removed from the axial tips of one of the coral fragments using stainless-steel bone-cutting forceps. Nubbins were extracted using the Bligh and Dyer method as described above, although the volume of solvents per 1 mg of extracted sample were adjusted (while maintaining the solvent ratios) in order to achieve a large enough volume so that the nubbin could be completely submersed in solvent (8700 µL total solvent volume). In order to accommodate the relatively large extraction solvent volume required for the nubbin samples, extractions for these samples were carried out in 50-mL polypropylene centrifuge tubes rather than 1.5-mL microcentrifuge tubes and glass vials.

#### 3.5.2. Tissue Powder Method 

A firm-bristled brush was used to remove powdered soft tissue from the surface of the other coral fragment. Each area of the fragment was brushed for approximately 10 s, with a goal of removing all soft tissue accessible and brushing each area of the fragment a single time. The removed powder was then collected into a homogeneous powder and aliquoted into three technical replicates (20 mg (±0.6 mg)) for extraction. Tissue powders were extracted using the Bligh and Dyer method as described in the extraction method comparison, with a total solvent volume of 1160 µL.

### 3.6. ^1^H NMR Spectroscopy Data Collection

Extracted, dried metabolites were rehydrated in 600 µL of deuterium-oxide-based 0.1 M sodium phosphate buffer containing 1 mM 3-trimethylsilylpropionic-2,2,3,3-d acid sodium salt (TMSP) as a chemical shift reference. Samples (550 µL) were then transferred to 5-mm NMR tubes (NORELL, Inc., Morganton, NC, USA) for ^1^H NMR analysis. All NMR profiles were obtained using a 700-MHz Bruker NMR spectrometer equipped with a TCI cryoprobe and Samplejet. One-dimensional ^1^H NMR spectra were acquired (Topspin version 3.2) using a nuclear Overhauser effect spectroscopy (NOESY) pulse sequence consisting of eight dummy scans and 160 scans for 65,536 data points with a relaxation delay of 3 s. Spectra were produced from the acquired FID via Fourier transformation. All spectra were referenced, phased, and baseline-corrected automatically (Topspin version 3.2).

### 3.7. Statistical Analysis

The different sample preparation methods were primarily assessed statistically using PCA alongside spectral %RSD, a benchmark metric for reproducibility in metabolomics [57]. To enable PCA analyses, the ^1^H NMR spectra from 0.5–10 ppm were divided into intelligent bins (NMR ProcFlow v1.2, [56]) with a signal-to-noise threshold equal to 3:1. Residual water (4.7–5.0 ppm) and other contaminants detected in the blank sample spectra (Appendix A, Appendix A) were manually excluded from binning. Binned spectra were normalized to constant sum and Pareto-scaled prior to multivariate analyses (MetaboAnalyst 4.0, [62]) in order to decrease the dominance of bins with the largest intensities [63]. PCAs were used to visually assess the quality of the data and to evaluate inter- and intra-method trends in the data.

Additionally, as a metric to quantifiably compare spectral intensity and coverage, spectra were once again intelligently binned as described above. However, here, samples of each treatment were binned separately rather than being grouped and binned by experiment. The number of features identified by the binning procedure was used to compare the spectral coverage of methods, and the total intensity from all detected features was averaged in order to compare spectral intensity of methods. 

Spectral %RSD values (%RSD = standard deviation/mean × 100; calculated for each bin, then the median of those values is reported as spectral %RSD) were used in order to evaluate the analytical reproducibility of the different methods. To do so, the ^1^H NMR spectra from 0.2–10 ppm were binned into 0.005-ppm increments resulting in 1960 bins (AMIX version 15.3, Bruker, Billerica, MA, USA). Residual water (4.7–5.0) and other contaminants detected in the blank sample spectra (Appendix A, Appendix A) were excluded from analysis. Binned spectra were normalized to constant sum and mean-centered.

### 3.8. Validation of Recommended Sample Preparation Workflow for Stony Corals

The methods from our recommended workflow were validated for use on experimental coral samples by assessing the ability to distinguish between different treatments. The growth anomaly and unaffected *P. compressa* samples described above were used to accomplish this. Tissue powder technical replicates were collected from the single lyophilized growth anomaly (*n* = 3) and unaffected (*n* = 3) samples, and were extracted using the Bligh and Dyer method. ^1^H NMR spectra from these samples were acquired with parameters described above, and were used to compare spectral coverage and intensity between treatment groups. PCA was also used to further assess the grouping of the two sample types. The spectra from the unaffected sample were also compared to *P. compressa*
^1^H NMR spectra from the literature [35] in order to further validate the selected workflow. Data for this comparison were obtained via an open-access metabolomics data repository (MetaboLights, [64]).

## 4. Conclusions

The current study represents a sizeable expansion on the previously published comparisons of sample preparation methods for reef-building coral samples destined for ^1^H-NMR-based metabolomics analysis. Specifically, our results provide a reference point that will be useful for selecting appropriate sample preparation methods in future coral metabolomics studies, and offers a recommended workflow that may help standardize sample preparation protocols in the field. The standardized use of our selected workflow would not only increase comparability across studies, but may also increase the explanatory power of metabolomics results, as it is specifically optimized for the investigation of stony coral systems.

## Figures and Tables

**Figure 1 metabolites-09-00032-f001:**
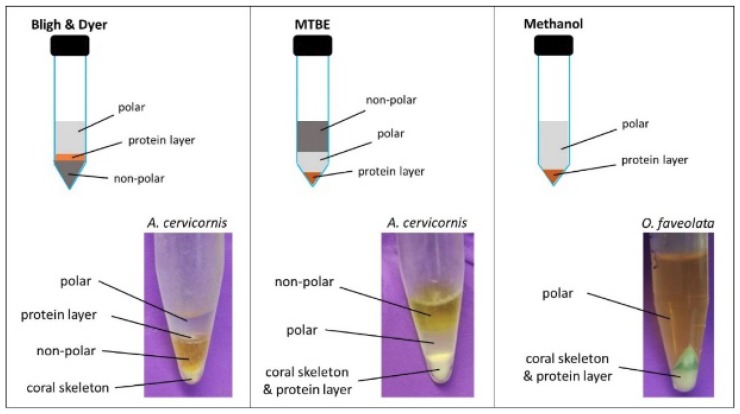
Diagram of solvent systems from three extraction methods: Bligh and Dyer, methyl *tert*-butyl ether (MTBE), and methanol extraction. *A. cervicornis* = *Acropora cervicornis; O. faveolata* = *Orbicella faveolata*.

**Figure 2 metabolites-09-00032-f002:**
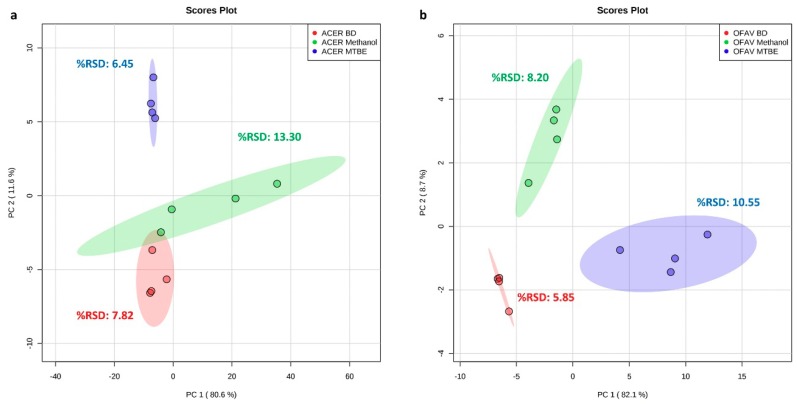
Principal component analysis (PCA) score plot and spectral relative standard deviation (%RSD) of three extraction methods (*n* = 4) for frozen homogenates of (**a**) *Acropora cervicornis* (ACER) and (**b**) *Orbicella faveolata* (OFAV). Ellipses indicate the 95% confidence region for each extraction method. BD = Bligh and Dyer extraction; Methanol = methanol extraction; MTBE = methyl *tert*-butyl ether extraction.

**Figure 3 metabolites-09-00032-f003:**
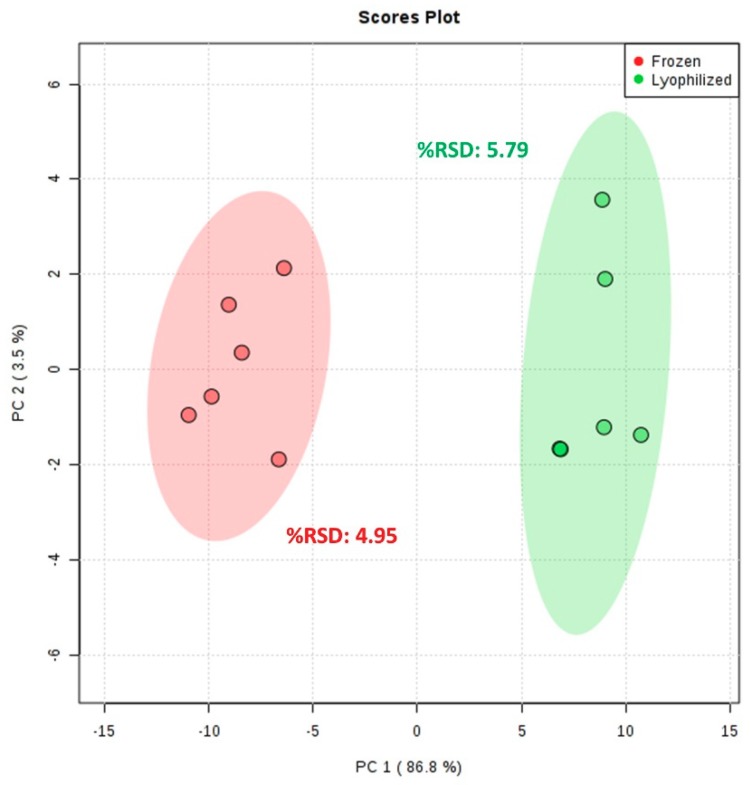
PCA score plot and spectral %RSD of frozen (*n* = 6) and lyophilized (*n* = 6) *Orbicella faveolata* homogenates. Metabolites were extracted using the Bligh and Dyer extraction method. Ellipses indicate the 95% confidence region for each metabolism preservation method.

**Figure 4 metabolites-09-00032-f004:**
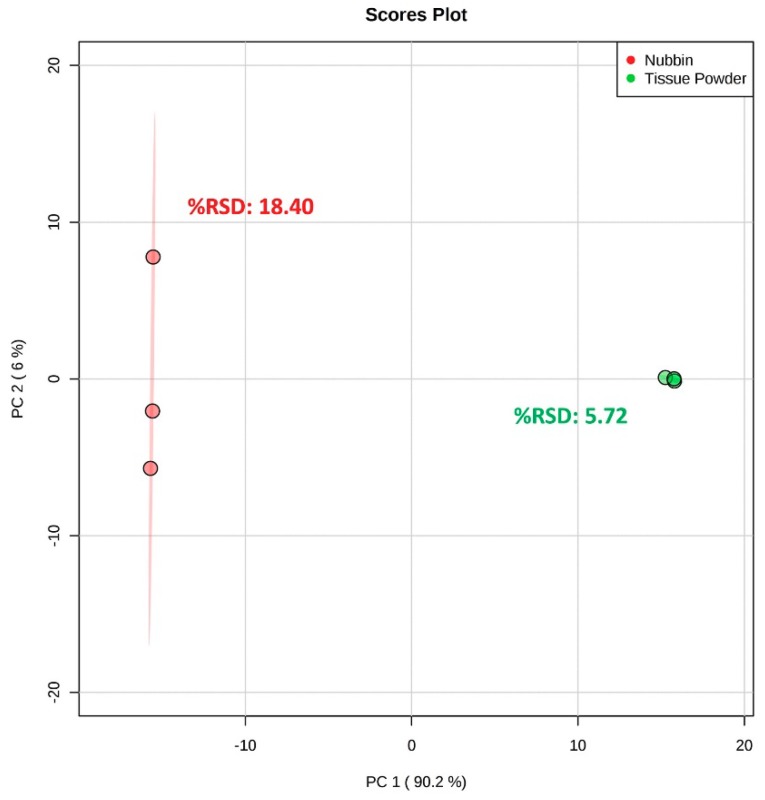
PCA score plot and spectral %RSD values comparing nubbin (*n* = 3) and tissue powder (*n* = 3) subsampling methods for unaffected *Porites compressa*. Samples were lyophilized and metabolites were extracted using the Bligh and Dyer extraction method. Ellipses indicate the 95% confidence region for each subsampling method.

**Figure 5 metabolites-09-00032-f005:**
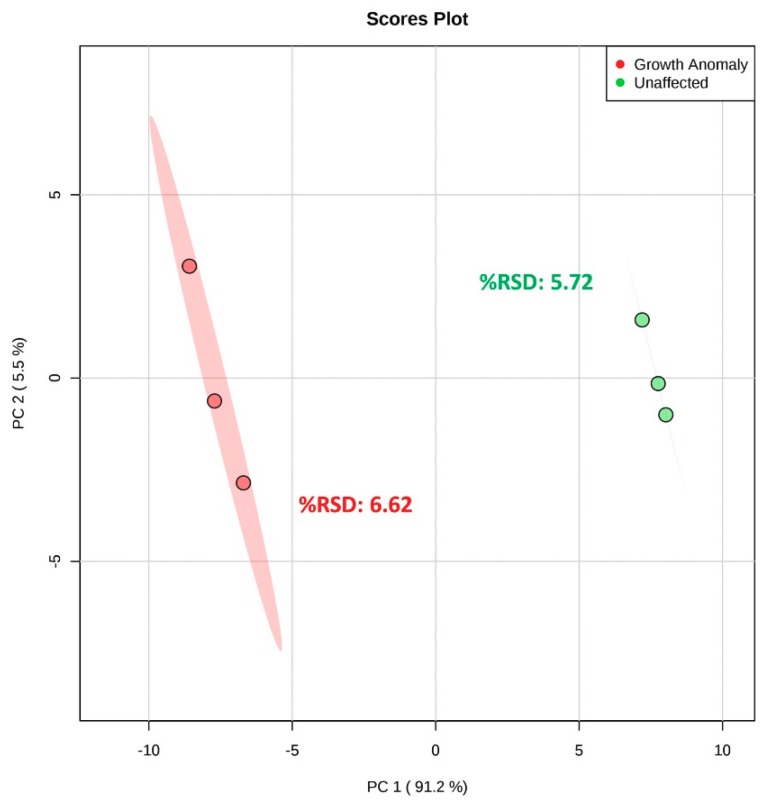
PCA score plot and spectral %RSD of the technical replicates from individual unaffected and growth anomaly samples. Data were collected according to the recommended workflow developed from the current study; samples were lyophilized, and subsampled using the tissue powder method, and metabolites were extracted using Bligh and Dyer extraction method.

**Figure 6 metabolites-09-00032-f006:**
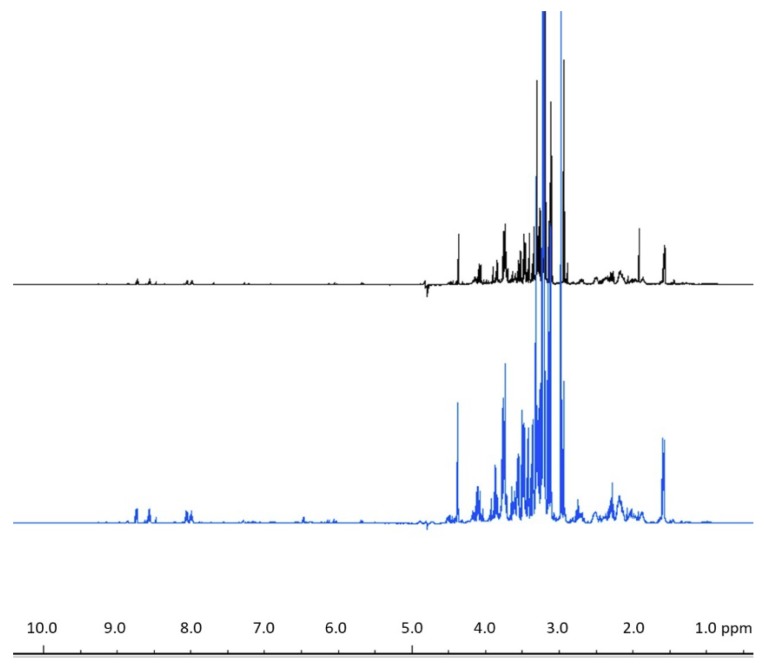
Representative ^1^H NMR spectrum of *Porites compressa* obtained using the recommended workflow from the current study (black) compared to a representative ^1^H NMR spectrum of *P. compressa* from the literature (blue) [35]. Spectra are normalized to chemical shift standard 3-trimethylsilylpropionic-2,2,3,3-d acid sodium salt (TMSP) at 0.0 ppm.

**Table 1 metabolites-09-00032-t001:** Number of features and average spectral intensities for the compared sample preparation methods calculated using an intelligent binning procedure on the raw spectra with a signal-to-noise threshold of 3:1 (NMR ProcFlow v1.2, [56]). MTBE = methyl *tert*-butyl ether.

Extraction	Species	Features Detected	Average Relative Intensity
Bligh and Dyer	*Acropora cervicornis*	248	1,289,537.264
MTBE	*Acropora cervicornis*	244	1,239,384.223
Methanol	*Acropora cervicornis*	332	2,037,465.296
Bligh and Dyer	*Orbicella faveolata*	301	3,619,779.736
MTBE	*Orbicella faveolata*	285	2,965,831.688
Methanol	*Orbicella faveolata*	305	4,303,333.548
Preservation			
Frozen	*Orbicella faveolata* *Orbicella faveolata*	340	3,453,549.791
Lyophilized		370	3,749,614.983
Subsampling			
Nubbin	*Porites compressa*	509	11,129,275.43
Tissue Powder	*Porites compressa*	342	1,811,468.673

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
