# Peer review of "Evaluation of Sample Preparation Methods for the Analysis of Reef-Building Corals Using 1H-NMR-Based Metabolomics"

_metabolites, 2019, doi:10.3390/metabo9020032_

Reviewer 1 Report

The manuscript aims to provide a valid protocol for the preparation of reef-building coral samples for NMR-based metabolomics analysis. To this aim, authors compared different methods for three important steps of the entire workflow, highlighting the relative benefits. The recommended workflow was also validated with a stony coral. Overall, the topic of the present paper is of high interest and utility for the scientific community, and confirms metabolomics as a powerful tool to be applied also on corals for biomonitoring activities. The manuscript is well structured and written, and discussion is good. However, introduction and background need to be enriched and supported by more recent updated literature, especially about the urgency of monitoring coral reef ecosystems and about the potential of metabolomics in the field of aquatic ecotoxicology.

Lines 34-47, in support of this section, authors should also provide evidences that the evaluation of the health status of sensitive coral reef ecosystems is a current and serious concern worldwide. Therefore, it is highly suggested to refer here also to very recent papers dealing on this, such as those of Ranjbar Jafarabadi et al. 2018, Mar. Pollut. Bull. 137, 185-197 (doi.org/10.1016/j.marpolbul.2018.09.052), Ranjbar Jafarabadi et al. 2018, Ecotoxicol. Environ. Saf. 163, 307-322 (doi.org/10.1016/j.ecoenv.2018.07.056), Ranjbar Jafarabadi et al. 2018, Si. Total Environ. 640-641, 1500-1511 (doi.org/10.1016/j.scitotenv.2018.06.029), and Ranjbar Jafarabadi et al. 2019, Chemosphere 215, 835-845 (doi.org/10.1016/j.chemosphere.2018.10.092).

Lines 58-59, for those not familiar with the potential of metabolomics, mainly in the field of aquatic ecology and ecotoxicology, more examples of its powerful use in elucidating organism response to various disturbance should be mentioned. For example, very recent studies successfully applied metabolomics in food safety (Cappello et al. 2018, Food Chem. Toxicol. 115, 391-397), as well as to unravel the mechanistic effects of metals (Cappello et al. 2016, Environ. Pollut. 219, 139-148), pharmaceuticals (Cappello et al. 2017, Ecotoxicol. Environ. Saf. 143, 166-172), and nanoparticles (Cappello et al. 2017, Comp. Biochem. Physiol. C 199, 20-27). Also, it should be mentioned and supported by literature the suitability of metabolomics to be applied in a wide range of aquatic organisms, from fish to invertebrates.

In Material and Methods section, please specify how many samples were analyzed for each method evaluated.

Author Response

First off, we would like to thank the reviewers for their thoughtful comments and edits to our manuscript. The reviewers points are well taken, and we have addressed their comments to the best of our ability. All line numbers indicated below are simple markup line numbers.

Point 1: Lines 34-47, in support of this section, authors should also provide evidences that the evaluation of the health status of sensitive coral reef ecosystems is a current and serious concern worldwide. Therefore, it is highly suggested to refer here also to very recent papers dealing on this, such as those of Ranjbar Jafarabadi et al. 2018, Mar. Pollut. Bull. 137, 185-197 (doi.org/10.1016/j.marpolbul.2018.09.052), Ranjbar Jafarabadi et al. 2018, Ecotoxicol. Environ. Saf. 163, 307-322 (doi.org/10.1016/j.ecoenv.2018.07.056), Ranjbar Jafarabadi et al. 2018, Si. Total Environ. 640-641, 1500-1511 (doi.org/10.1016/j.scitotenv.2018.06.029), and Ranjbar Jafarabadi et al. 2019, Chemosphere 215, 835-845 (doi.org/10.1016/j.chemosphere.2018.10.092).

Response 1: To address this point, language was added to indicate that evaluation of coral reef ecosystems is a current and serious issue worldwide (lines 47-49). In support of this assertion, we have also provided four new references: Hughes et al. 2018, Ranjbar Jafarabadi et al. 2018, Glynn et al. 2018, and Sawaya et al. 2018 (line 49).

Point 2: Lines 58-59, for those not familiar with the potential of metabolomics, mainly in the field of aquatic ecology and ecotoxicology, more examples of its powerful use in elucidating organism response to various disturbance should be mentioned. For example, very recent studies successfully applied metabolomics in food safety (Cappello et al. 2018, Food Chem. Toxicol. 115, 391-397), as well as to unravel the mechanistic effects of metals (Cappello et al. 2016, Environ. Pollut. 219, 139-148), pharmaceuticals (Cappello et al. 2017, Ecotoxicol. Environ. Saf. 143, 166-172), and nanoparticles (Cappello et al. 2017, Comp. Biochem. Physiol. C 199, 20-27). Also, it should be mentioned and supported by literature the suitability of metabolomics to be applied in a wide range of aquatic organisms, from fish to invertebrates.

Response 2: To address this point, language was added to further clarify the potential of metabolomics in a range of fields, including aquatic ecotoxicology (lines 65-67). In support of this point, three additional references are included as examples of the utility of metabolomics in these fields: Watanabe et al. 2015, Cappello et al. 2017, and Casu et al. 2017 (line 67).

Additionally, we inserted a mention that  metabolomics is suitable to be applied to a wide range of aquatic organisms (lines 64-65), and have included three new references as examples: Southam et al. 2008, Boroujerdi et al. 2009, Schock et al. 2010 (line 65).

Point 3: In Material and Methods section, please specify how many samples were analyzed for each method evaluated.

Response 3: The number of samples analyzed are indicated in text as “(n= )” for the extraction method comparison (lines 450-451), the metabolism preservation comparison (line 493), the subsampling method comparison (line 507) and the workflow validation (lines 565-566). Further clarification was added for the number of samples analyzed during the extraction method comparison (lines 450-451).

Reviewer 2 Report

This paper evaluates sample preparation methods for analysis of reef-building corals. As such, topic is timely as it is imminent to gather in-depth understanding of coral processes and functioning. However, author should address the following comments in the manuscript in relevant and appropriate sections:

1) Authors have used two species for testing the methods and then validated with one environmentally relevant coral species. Can we extrapolate the results from these three species to all the other species which are of relevance? Given the heterogeneity of environmental corals. To state that it is a proposed workflow, authors have to consider including other species as well.

2) Authors mentioned about capturing coral holobiont and how it is important. Agreed, it is quite relevant as they do not function in isolation. However, given the complexity of coral metabolome, how do you differentiate between host metabolites vs shared metabolites vs microbial metabolites?

3) A large proportion of figures presented here are PCA plots, authors can use a different way to present their data for readers benefits.

4) Lastly, Authors mentioned in the beginning about studies done in past reporting extraction methods for these corals, it would be good to include a summary table to findings such as number of features etc from those study, vs. this study for reader's reference.

Author Response

First off, we would like to thank the reviewers for their thoughtful comments and edits to our manuscript. The reviewers points are well taken, and we have addressed their comments to the best of our ability. All line numbers indicated below are simple markup line numbers.

Point 1: Authors have used two species for testing the methods and then validated with one environmentally relevant coral species. Can we extrapolate the results from these three species to all the other species which are of relevance? Given the heterogeneity of environmental corals. To state that it is a proposed workflow, authors have to consider including other species as well.

Response 1: We have added new language to caution against blind extrapolation of our results since only three different coral species were used in our study (lines 403-405). However, we also noted that the three species that we used span a range of genera, as well as skeletal and morphological characteristics, so our results may be more widely applicable than the number of species tested may indicate (lines 405-407). However, because every individual metabolomics workflow will require some level of optimization, we continue to believe that our recommended workflow provides a useful reference point for all environmentally relevant coral species and may help standardize sample preparation protocols in the field of coral metabolomics.

Point 2: Authors mentioned about capturing coral holobiont and how it is important. Agreed, it is quite relevant as they do not function in isolation. However, given the complexity of coral metabolome, how do you differentiate between host metabolites vs shared metabolites vs microbial metabolites?

Response 2: At the current time, our methods do not allow us to differentiate between host metabolites vs metabolites from other parts of the holobiont. We have accordingly added new language to reflect this (lines 86-87 and lines 351-354). However, we have also mentioned that pairing metabolomics data with complementary –omics data may help elucidate where specific metabolites originate from within the holobiont (lines 352-354).

Point 3:  A large proportion of figures presented here are PCA plots, authors can use a different way to present their data for readers benefits.

Response 3: We believe that the given PCA plots (Figures 2-5) with accompanying spectral %RSD values provide the best visualization of the quality and reproducibility. Additionally, two of our figures, a diagram of the different extraction methods (Figure 1) and representative 1H NMR spectra from the current study compared to the literature (Figure 6), provide alternative and complementary methods we use in our analysis. Furthermore, for the readers benefits we have included five supplementary figures which display representative 1H NMR spectra from each of the three method comparison within our study (Figure S1-S4), as well as a conceptual diagram of the nubbin and tissue powder subsampling methods (Figure S5).

Point 4: Lastly, Authors mentioned in the beginning about studies done in past reporting extraction methods for these corals, it would be good to include a summary table to findings such as number of features etc from those study, vs. this study for reader's reference.

Response 4: We originally included a summary table detailing the different sample preparation methods used by each of these studies, which we have bolstered by adding an additional column indicating which coral species were included in each study (Table S1). In general, we hesitate to add comparison of findings (such as number of features) from these studies since the methods vary wildly between studies. For example, we do not find much utility in comparing number of features between NMR-based and MS-based studies, nor comparing the spectral intensities from studies using different NMR magnet strengths. However, we have included an additional comparison of spectral %RSD between our study and a study (Sogin et al. 2014) we consider to be comparable (lines 335-338). To further detail this comparison, we have also included an additional table in the supplementary materials (Table S4).
